# Predicting Milling Stability Based on Composite Cotes-Based and Simpson’s 3/8-Based Methods

**DOI:** 10.3390/mi13050810

**Published:** 2022-05-23

**Authors:** Xu Du, Pengfei Ren, Junqiang Zheng

**Affiliations:** 1Mechatronic Institute, Zhejiang Sci-Tech University, Hangzhou 310018, China; duxu@zstu.edu.cn (X.D.); renpengfei596@gmail.com (P.R.); 2School of Mechanical Engineering, Hangzhou Dianzi University, Hangzhou 310018, China

**Keywords:** milling stability, composite Cotes-based method, Simpson’s 3/8-based method, Floquet theory

## Abstract

Avoiding chatter in milling processes is critical for obtaining machined parts with high surface quality. In this paper, we propose two methods for predicting the milling stability based on the composite Cotes and Simpson’s 3/8 formulas. First, a time-delay differential equation is established, wherein the regenerative effects are considered. Subsequently, it is discretized into a series of integral equations. Based on these integral equations, a transition matrix is determined using the composite Cotes formula. Finally, the system stability is analyzed according to the Floquet theory to obtain the milling stability lobe diagrams. The simulation results demonstrate that for the single degree of freedom (single-DOF) model, the convergence speed of the composite Cotes-based method is higher than that of the semi-discrete method and the Simpson’s equation method. In addition, the composite Cotes-based method demonstrates high computational efficiency. Moreover, to further improve the convergence speed, a second method based on the Simpson’s 3/8 formula is proposed. The simulation results show that the Simpson’s 3/8-based method has the fastest convergence speed when the radial immersion ratio is large; for the two degrees of freedom (two-DOF) model, it performs better in terms of calculation accuracy and efficiency.

## 1. Introduction

The three main types of vibration that occur during high-speed milling are free vibration, excited vibration, and self-excited vibration, and the regenerative chatter in self-excited vibration is the main cause of instability in the machining process [1]. Because this chatter typically leads to poor surface quality of the machined parts, aggravated tool wear, and even reduced service life of machine tools, it must be avoided to solve or overcome these problems [2]. In the analysis of chatter, dynamic milling processes that consider the delay effect are generally described using delay differential equations with respect to the time-periodic coefficients [3,4]. The chatter stability based on these differential equations is an important index for achieving high-performance machining [5].

In recent decades, experimental, analytical, and numerical methods have been studied to obtain stability lobe diagrams, which can be used to determine exact values avoiding chatter. Wu et al. [6] utilized the Lyapunov index to measure whether chatter occurred; however, they could only predict the system stability under specific parameters. Davies et al. [7] used the time-domain calculation method to predict unstable regions in the stability lobe diagrams, which was only applicable to small radial depths of cut. Altintas and Budak et al. [8] first presented a zero-order approximation method to quickly obtain the stability lobe diagrams, where the real and imaginary parts of the feature are used to determine the cutting parameters. It is worth mentioning that this method was only applicable to high radial depths of cut. Bayly et al. [9] employed the time-finite element method to calculate tool motion and utilized the weighted residual method to determine the Floquet transition matrix. Based on the delay differential equations, Insperger et al. [10] proposed the semi-discretization method (SDM) and the first-order semi-discretization method (1_st_SDM), which are widely used as the basis for evaluating other stability analysis methods in the time domain. The SDM and 1_st_SDM only discretized delay states and periodic coefficients. Based on direct integration, Ding et al. [11] presented a full-discretization method (FDM) to predict milling stability. Unlike the SDM and 1_st_SDM, the FDM carried out the linear interpolation on the state terms to improve computational efficiency. Further, Insperger [12] compared the FDM with the SDM and 1_st_SDM in terms of the convergence speed and computational efficiency. It was found that the convergence speed of the 1_st_SDM was faster than those of the SDM and FDM, while the computational efficiency of the FDM was higher than those of the SDM and 1_st_SDM. After the comparisons, some FDM-based methods were used to predict the milling stability. In addition to the above methods, the numerical integration methods are widely utilized to obtain the stability lobe diagrams. Ding et al. [13] approximated the integral terms with the Newton–Cotes and Gauss integral formulas to predict the milling stability. Lu et al. [14] approximated the solution of the delay differential equation with the direct integration technique. Qin et al. [15] approximated the state terms of the delay differential equation with the Chebyshev wavelets of the second kind. Moreover, Liu et al. [16] combined the numerical solution of the delay differential equation with the Simpson formula to predict milling stability (SEM), which was only suitable for large radial immersion. However, the aforementioned numerical methods cannot simultaneously guarantee fast convergence speed and high computational efficiency.

To this end, we present two numerical analysis methods, namely the composite Cotes-based method (CCM) and Simpson’s 3/8-based method (S38M), in this paper to improve the convergence speed and computational efficiency at the same time. The remainder of this paper is organized as follows. Section 2 presents the mathematical model of system motion with two degrees of freedom (DOF) and the state-space. The composite Cortes-based method and its simulation and analysis for the single-DOF model are presented in Section 3. Subsequently, the Simpson’s 3/8-based method and its simulation and analysis for the single-DOF and two-DOF models are presented in Section 4. Finally, the conclusions are stated in Section 5.

## 2. Mathematical Model

Taking down-milling as an example, the dynamic system of end-milling with two degrees of freedom is shown in Figure 1, where the workpiece is assumed to be rigid, and the milling cutter is assumed to be evenly distributed. Note that the helix angle of the milling cutter is not taken into account. Considering the regenerative effect, the mathematical model of system motion can be expressed as a second-order differential equation [17] as follows:(1)$$M\ddot{q}\left(t\right)+C\dot{q}\left(t\right)+Kq\left(t\right)=-a_{p}K_{c}\left(t\right)[q\left(t\right)-q\left(t-T\right)]$$
where q(t), $\dot{q}\left(t\right)$, and $\ddot{q}\left(t\right)$ represent the displacement vector, the first derivative of **q**(*t*) w.r.t *t*, and the second derivative of **q**(*t*) w.r.t *t*, respectively. **M**, **C**, **K**, and *a_p_* represent the mass matrix, damping matrix, stiffness matrix, and depth of cut, respectively. *T* = 60/(*Nv*) represents the delay period of the delay differential equations, where *N* represents the cutter tooth number, and *v* represents the spindle speed. The radial cutting force coefficient matrix **K***_c_*(*t*) can be expressed as follows:(2)$$K_{c}\left(t\right)=\left[\begin{array}{cc} h_{xx}\left(t\right) & h_{xy}\left(t\right)\\ h_{yx}\left(t\right) & h_{yy}\left(t\right) \end{array}\right]$$
where
(3)$$h_{xx}\left(t\right)=\sum_{j=1}^{N}g\left(\phi_{j}\left(t\right)\right)\sin\left(\phi_{j}\left(t\right)\right)\left[K_{t}\cos\left(\phi_{j}\left(t\right)\right)+K_{n}\sin\left(\phi_{j}\left(t\right)\right)\right]$$
(4)$$h_{xy}\left(t\right)=\sum_{j=1}^{N}g\left(\phi_{j}\left(t\right)\right)\cos\left(\phi_{j}\left(t\right)\right)\left[K_{t}\cos\left(\phi_{j}\left(t\right)\right)+K_{n}\sin\left(\phi_{j}(t)\right)\right]$$
(5)$$h_{yx}\left(t\right)=\sum_{j=1}^{N}g\left(\phi_{j}\left(t\right)\right)\sin\left(\phi_{j}\left(t\right)\right)\left[-K_{t}\sin\left(\phi_{j}\left(t\right)\right)+K_{n}\cos\left(\phi_{j}(t)\right)\right]$$
and
(6)$$h_{yy}\left(t\right)=\sum_{j=1}^{N}g\left(\phi_{j}\left(t\right)\right)\cos\left(\phi_{j}\left(t\right)\right)\left[-K_{t}\sin\left(\phi_{j}\left(t\right)\right)+K_{n}\cos\left(\phi_{j}(t)\right)\right]$$
where *K_t_* and *K_n_* represent the tangential and normal cutting force coefficients, respectively. $\phi_{j}\left(t\right)$ represents the angular position of tooth *j,* as shown in Figure 1, and is defined as follows:(7)$$\phi_{j}\left(t\right)=\frac{2\pi v}{60}t+\frac{2\pi \left(j-1\right)}{N}$$

$g\left(\phi_{j}\left(t\right)\right)$ is a piecewise function used to judge whether the tooth *j* is cutting or not and is defined as follows:(8)$$g\left(\phi_{j}\left(t\right)\right)=\{\begin{array}{ll} 1 & \phi_{st}<\phi_{j}\left(t\right)<\phi_{ex}\\ 0 & \mathrm{otherwise} \end{array}$$
where $\phi_{st}$ and $\phi_{ex}$ represent the start and exit cutting angles of the tooth *j*, respectively. In the up-milling process, $\phi_{st}=0$ and $\phi_{ex}=\arccos\left(1-2a/D\right)$; in the down-milling process, $\phi_{st}=\arccos(2a/D-1)$ and $\phi_{ex}=\pi$, where *a*/*D* represents the radial immersion ratio.

## 3. Composite Cotes-Based Method

### 3.1. State-Space Expression

**x**(*t*) can be defined as $x\left(t\right)=\left[\begin{array}{c} q\left(t\right)\\ M\dot{q}\left(t\right)+Cq\left(t\right)/2 \end{array}\right]$. Then, the state expression of (1) can be expressed as follows:(9)$$\dot{x}\left(t\right)=Ax\left(t\right)+a_{p}B\left(t\right)\left[x\left(t\right)-x\left(t-T\right)\right]$$
where
(10)$$A=\left[\begin{array}{cc} -M^{-1}C/2 & M^{-1}\\ CM^{-1}C/4-K & -CM^{-1}/2 \end{array}\right]$$
(11)$$B(t)=\left[\begin{array}{cc} 0 & 0\\ K_{c}(t) & 0 \end{array}\right]$$

If $a_{p}B\left(t\right)\left[x\left(t\right)-x\left(t-T\right)\right]$ is regarded as the inhomogeneous term of $\dot{x}\left(t\right)=Ax\left(t\right)$, then the following equation can be derived:(12)$$x\left(t\right)=e^{A\left(t-t_{0}\right)}x\left(t_{0}\right)+a_{p}\int_{t_{0}}^{t}\left\{e^{A\left(t-\xi \right)}B\left(\xi \right)\left[x\left(\xi \right)-x\left(\xi -T\right)\right]\right\}d\xi$$
where *t*_0_ represents the start time. It is worth noting that **x**(*ξ*) is unknown such that (12) is a Volterra integral equation of the second kind.

### 3.2. Numerical Algorithm

To solve (9) using the composite Cotes formula, the integral equation *f*(*x*) is assumed to be continuous in the interval [*c*, *d*]. Then, [*c*, *d*] is divided into *m* isometric intervals; that is, the span of each interval *h* is equal to (*d − c*)/*m*. Based on the isometric nodes *u_k_* = *c* + (*k* − 1)/*h*, where *k* = 1, 2, …, *m* + 1, we obtain the following:(13)$$I_{m}=\left(d-c\right)\sum_{k=0}^{m}C_{k}^{\left(m\right)}f\left(u_{k}\right)$$
where $C_{k}^{\left(m\right)}$ represents the Cotes coefficients [18], expressed as follows:(14)$$C_{k}^{\left(m\right)}=\frac{{\left(-1\right)}^{m-k}}{nk!\left(n-k\right)}\int_{0}^{m}\prod_{j=0,j\neq k}^{m}\left(t-j\right)dt,k=0,1,\ldots ,m$$

It is worth noting that $C_{k}^{\left(m\right)}$ only depends on *m*. When *m* = 4, (14) can be expressed as follows:(15)$$C=\frac{1}{90}\left[7f\left(u_{0}\right)+32f\left(u_{1}\right)+12f\left(u_{2}\right)+32f\left(u_{3}\right)+7f\left(u_{4}\right)\right]$$
where *T* consists of the durations of the free and excited vibrations [15]. The durations of the free vibration and excited vibration can be defined as *t_f_* and *t_c_*, respectively. For the free vibration, **B**(*ξ*) = 0, and (12) can be expressed as follows:(16)$$x\left(t\right)=e^{A\left(t-t_{0}\right)}x\left(t_{0}\right)$$

Therefore, the state equation can be solved when *t = t*_0_ + *t_f_*, and it can be expressed as follows:(17)$$x\left(t_{0}+t_{f}\right)=e^{At_{f}}x\left(t_{0}\right)$$

For the excited vibration, its time lies in [*t*_0_ + *t_f_*, *t*_0_ + *T*]. The interval is divided into *n* isometric intervals, and the span of each interval *l* is equal to (*T − t_f_*)/*n*. The discrete time can be expressed as *t_i_* = *t*_0_ + *t_f_* + (*i* − 1)*h*, where *i* = 1, 2, …, *n* + 1. According to the numerical integral solution of the Volterra integral equation of the second kind [19], we obtain the following:(18)$$x\left(t_{i+1}\right)=e^{A\left(t_{i+1}-t_{i}\right)}x\left(t_{i}\right)+a_{p}\int_{t_{i}}^{t_{i+1}}\left\{e^{A\left(t_{i+1}-\xi \right)}B\left(\xi \right)\left[x\left(\xi \right)-x\left(\xi -t\right)\right]\right\}d\xi$$

If [*t_i_*, *t_i_*
_+ 1_] is divided into four isometric intervals, then *t_i_*, *t_i+_*_1/4_, *t_i+_*_1/2_, *t_i+_*_3/4_, and *t_i+_*_1_ can be derived. Combining (13) and (15), the integral equation can be expressed as follows:(19)$$\int_{t_{i}}^{t_{i+1}}f\left(\xi \right)d\xi =\frac{t_{i+1}-t_{i}}{90}\left[7f\left(t_{i}\right)+32f\left(t_{i+1/4}\right)+12f\left(t_{i+1/2}\right)+32f\left(t_{i+3/4}\right)+7f\left(t_{i+1}\right)\right]$$

Upon substituting (19) into (18), we can see that **x**(*t_i_*_+1/4_), **x**(*t_i_*_+1/2_), and **x**(*t_i_*_+3/4_) cannot be solved directly. The barycentric Lagrange interpolation method [20] is introduced here to approximate **x**(*t_i_*_+1/4_), **x**(*t_i_*_+1/2_), and **x**(*t_i_*_+3/4_) and obtain the following:(20)$$x\left(t_{i+1/4}\right)≃\frac{21x\left(t_{i}\right)+14x\left(t_{i+1}\right)-3x\left(t_{i+2}\right)}{32}$$
(21)$$x\left(t_{i+1/2}\right)≃\frac{3x\left(t_{i}\right)+6x\left(t_{i+1}\right)-x\left(t_{i+2}\right)}{8}$$
(22)$$x\left(t_{i+3/4}\right)≃\frac{5x\left(t_{i}\right)+30x\left(t_{i+1}\right)-3x\left(t_{i+2}\right)}{32}$$

Substituting (19)–(22) into (18), **x**(*t_i_*) can be expressed as follows:(23)$$\begin{array}{ll} x\left(t_{i+1}\right) & =e^{Ah}x\left(t_{i}\right)+\frac{a_{p}h}{6}\{\frac{5}{2}e^{Ah}B\left(t_{i}\right)\left(x\left(t_{i}\right)-x\left(t_{i}-T\right)\right)+4B\left(t_{i+1}\right)\left(x\left(t_{i+1}\right)-x\left(t_{i+1}-T\right)\right)\\ & -\frac{1}{2}e^{-Ah}B\left(t_{i+2}\right)\left(x\left(t_{i+2}\right)-x\left(t_{i+2}-T\right)\right)\} \end{array}$$

It is worth noting that **B**(*t_n_*_+2_) cannot be calculated directly using (9) when *i* = *n*. A Newton Cotes formula [13] is introduced here to calculate **x**(*t_n_*_+1_) and is expressed as follows:(24)$$x\left(t_{n+1}\right)=e^{Ah}x\left(t_{n}\right)+\frac{a_{p}h}{2}\left[e^{Ah}B\left(t_{n}\right)\left(x\left(t_{n}\right)-x\left(t_{n}-T\right)\right)+B\left(t_{n+1}\right)\left(x\left(t_{n+1}\right)-x\left(t_{n+1}-T\right)\right)\right]$$

Combining (17), (23), and (24), the transition matrix can be constructed as follows:(25)$$C_{1}={\left[\begin{array}{ccccccc} 0 & 0 & 0 & \cdots & 0 & 0 & 0\\ -e^{Ah} & 0 & 0 & \cdots & 0 & 0 & 0\\ 0 & -e^{Ah} & 0 & \cdots & 0 & 0 & 0\\ \vdots & \vdots & \vdots & \ddots & \vdots & \vdots & \vdots \\ 0 & 0 & 0 & \cdots & 0 & 0 & 0\\ 0 & 0 & 0 & \cdots & -e^{Ah} & 0 & 0\\ 0 & 0 & 0 & \cdots & 0 & -e^{Ah} & 0 \end{array}\right]}_{\left(2n+2\right)\times \left(2n+2\right)}$$
(26)$$D_{1}={\left[\begin{array}{ccccccc} 0 & 0 & 0 & \cdots & 0 & 0 & 0\\ -\frac{75}{2}e^{Ah}B_{1} & -60B_{2} & \frac{15}{2}e^{-Ah}B_{3} & \cdots & 0 & 0 & 0\\ 0 & -\frac{75}{2}e^{Ah}B_{2} & -60B_{3} & \cdots & 0 & 0 & 0\\ \vdots & \vdots & \vdots & \ddots & \vdots & \vdots & \vdots \\ 0 & 0 & 0 & \cdots & -60B_{i-1} & \frac{15}{2}e^{-Ah}B_{i} & 0\\ 0 & 0 & 0 & \cdots & -\frac{75}{2}e^{Ah}B_{i-1} & -60B_{i} & \frac{15}{2}e^{-Ah}B_{i+1}\\ 0 & 0 & 0 & \cdots & 0 & 45e^{Ah}B_{i} & 45B_{i+1} \end{array}\right]}_{\left(2n+2\right)\times \left(2n+2\right)}$$
(27)$$E_{1}={\left[\begin{array}{ccccc} 0 & 0 & \cdots & 0 & e^{At_{f}}\\ 0 & 0 & \cdots & 0 & 0\\ \vdots & \vdots & \ddots & \vdots & \vdots \\ 0 & 0 & \cdots & 0 & 0\\ 0 & 0 & \cdots & 0 & 0 \end{array}\right]}_{\left(2n+2\right)\times \left(2n+2\right)}$$
where **B***_i_* represents **B**(*t_i_*), *i* = 1, 2, …, *n* + 1.

According to (25)–(27), the dynamic mapping of the discrete system can be determined as follows:(28)$$\left(I_{1}+C_{1}+\frac{a_{p}h}{90}D_{1}\right)\left[\begin{array}{c} x\left(t_{1}\right)\\ x\left(t_{2}\right)\\ \vdots \\ x\left(t_{n+1}\right) \end{array}\right]=\left(\frac{a_{p}h}{90}D_{1}+E_{1}\right)\left[\begin{array}{c} x\left(t_{1}-T\right)\\ x\left(t_{2}-T\right)\\ \vdots \\ x\left(t_{n+1}-T\right) \end{array}\right]$$
where **I**_1_ is a (2*n* + 1) × (2*n* + 1) identity matrix.

Therefore, the transition matrix of the dynamic system in a tool pass cycle is obtained as follows:(29)$$\Phi_{1}={\left(I_{1}+C_{1}+\frac{a_{p}h}{90}D_{1}\right)}^{-1}\left(\frac{a_{p}h}{90}D_{1}+E_{1}\right)$$

It is worth noting that the chatter stability must be determined using the Floquet theory. If the modulus of any eigenvalue in **Φ**_1_ exceeds 1, the system is unstable; if the moduli of all eigenvalues in **Φ**_1_ are less than 1, the system is stable [21,22]. Therefore, the boundary curve between the unstable and stable regions in the lobe diagram can be used as the criterion for judging whether chatter occurs.

### 3.3. Simulation and Analysis

In this section, SDM [10] and SEM [16] are compared. For objective comparison, the CCM, SDM, and SEM share the same parameters and machining conditions. A benchmark example of the single-DOF milling model is utilized to validate and analyze the CCM. The single-DOF milling mathematical model is expressed as follows [23]:(30)$$\ddot{x}\left(t\right)+2\zeta \omega_{n}\dot{x}\left(t\right)+\omega_{n}^{2}x\left(t\right)=-\frac{a_{p}h\left(t\right)}{m_{t}}\left(x\left(t\right)-x\left(t-T\right)\right)$$
where *ζ* denotes the relative damping, *ω_n_* denotes the angular natural frequency, and *h*(*t*) denotes the specific cutting force coefficient.

The state-space for single-DOF milling mathematical model is expressed as follows:(31)$$\dot{x}\left(t\right)=Ax\left(t\right)+a_{p}B\left(t\right)\left[x\left(t\right)-x\left(t-T\right)\right]$$
where
(32)$$A=\left[\begin{array}{cc} -\zeta \omega_{n} & \frac{1}{m_{t}}\\ m_{t}\omega_{n}^{2}\left(\zeta^{2}-1\right) & -\zeta \omega_{n} \end{array}\right]$$
(33)$$B\left(t\right)=\left[\begin{array}{cc} 0 & 0\\ -h\left(t\right) & 0 \end{array}\right]$$
where *h*(*t*) is equal to *h_xx_*(*t*), defined in (3).

The parameters in (32) and (33) are defined as follows: *f_n_* = 922 Hz, *ξ* = 0.011, *m_t_* = 0.3993 kg, *K_t_* = 6 × 10^8^ N/m^2^ and *K_t_* = 2 × 10^8^ N/m^2^. The adopted machining condition is down-milling. The radial immersion ratio *a*/*D* is set as 1 to avoid intermittent milling. The spindle speed is set as 5000 rpm (*v* = 5000 rpm). The depths of cut are set as 0.001 m, 0.0005 mm, and 0.0002 mm, respectively. All programs in this study are executed in MATLAB R2019a and run on a personal computer (AMD Ryzen 5 5600H; CPU 4.0 GHz, 16 GB). The maximum modulus of the eigenvalues of **Φ** is labelled as |λ|, and the maximum modulus of the eigenvalues of **Φ** with SDM is labelled as $\left|\lambda_{0}\right|$. In this study, $\left|\lambda_{0}\right|$ is treated as the exact value.

The convergence rate comparisons of the CCM, SDM, and SEM are shown in Figure 2. It is seen from the figure that when *n* is small, the convergence rate of the CCM is significantly higher than those of the SDM and SEM regardless of the depth of cut. As *n* increases, the convergence rates of the three methods approach gradually, and the convergence rates of the CCM and SEM are higher than that of SDM. The figure (a) is enlarged to observe the differences better. It is worth noting that the convergence rate of the CCM is significantly higher than those of the SDM and SEM when the discrete number changes from 75 to 100. The results demonstrate that the convergence rate of the CCM outweighs those of the SDM and SEM.

The stability lobe diagrams can be used to compare the calculation accuracy among the CCM, SDM, and SEM. As a reference, the discrete number is set as 500. The other parameters are set as follows: the discrete numbers are set as 25, 40, and 55; the radial immersion ratios are set as 0.05, 0.5, and 1; the spindle speed varies from 0.5 × 10^4^ to 2.5 × 10^4^ rpm, and 200 equally distributed sampling points are selected within this range; the depth of cut varies from 0 to 0.01 m, and 100 equally distributed sampling points are selected within this range.

In the single-DOF system, the stability lobe diagrams obtained with the CCM, SDM, and SEM when *a*/*D* = 1 are given in Table 1. It can be seen that the stability lobe diagrams of the CCM are closer to the reference ones than the SDM and SEM. Note that the severe distortions appear in the stability lobe diagrams obtained with the SEM when the discrete number is 25. Overall, the CCM is more accurate than the SDM and SEM. To further compare the calculation accuracy, two indicators are introduced, i.e., the arithmetic mean of relative error (*AMRE*) and mean squared error (*MSE*) [2]. The *AMRE* and *MSE* are defined as follows [2]:$$AMRE=\frac{1}{n}\sum_{i=1}^{n}\frac{\left|a_{i}-a_{i0}\right|}{a_{i0}}$$
$$MSE=\frac{1}{n}\sum_{i=1}^{n}{\left(a_{i}-a_{i0}\right)}^{2}$$
where *a_i_* and *a_i_*_0_ denote the predicted axial depth and reference axial depth, respectively. *n* denotes the discrete number. The discrete numbers are set as 25, 30, 35, 40, 45, 50, and 55. The variations of the *AMRE* and *MSE* are depicted in Figure 3. Due to the fact that the severe distortions appear in the stability lobe diagrams obtained with the SEM when the discrete number is 25, the *AMRE* and *MSE* of the SEM cannot be attained. Take *n* = 40 as an example. The *AMRE* obtained with the CCM, SDM, and SEM is 0.041, 0.076, and 0.154, respectively. The *MSE* obtained with the CCM, SDM, and SEM is 2.62 × 10^−8^, 4.24 × 10^−8^, and 1.07 × 10^−7^, respectively. The results show that the CCM is closer to the reference values than the SDM and SEM.

Furthermore, the computational efficiency is an important criterion for measuring the effectiveness of the chatter analysis methods. Figure 4 shows the computational time of the CCM, SDM, and SEM under different discrete numbers when *a*/*D* = 1. We observe that the computational time can be reduced from 55.63 s to 20.21 s, 62.45 s to 25.79 s, 72.06 s to 30.13 s, and 81.21 s to 35.18 s when the discrete numbers are 40, 45, 50, and 55, respectively. Furthermore, the stability lobe diagrams, *AMRE* and *MSE,* as well as the computational time when *a*/*D* = 0.05 and *a*/*D* = 0.5 are given in Table 2 and Table 3 and Figure 5 and Figure 6. It could be noted that the stability lobe diagrams obtained with the CCM are closer to the reference ones than those obtained with the SDM and SEM, and the CCM has the highest calculation accuracy. The results also demonstrate that the CCM has significant advantages in the computational efficiency.

## 4. Simpson’s 3/8-Based Method

### 4.1. State-Space Expression

To match with S38M, a new state-space expression is defined as follows [17]:(34)$$\dot{x}\left(t\right)=Ax\left(t\right)+a_{p}B\left(t\right)\left[x\left(t\right)-x\left(t-T\right)\right]$$
where
(35)$$A=\left[\begin{array}{cc} 0 & I\\ -M^{-1}K & -M^{-1}C \end{array}\right]$$
and
(36)$$B(t)=\left[\begin{array}{cc} 0 & 0\\ -M^{-1}K_{c}\left(t\right) & 0 \end{array}\right]$$

Based on the numerical integration method of the Volterra integral equation of the second kind [19], the state-space equation can be deduced as follows:(37)$$x\left(t\right)=e^{A\left(t-t_{st}\right)}x\left(t_{st}\right)+a_{p}\int_{t_{st}}^{t}\left\{e^{A\left(t-\xi \right)}B\left(\xi \right)\left[x\left(\xi \right)-x\left(\xi -T\right)\right]\right\}d\xi$$
where *t_st_* represents the start time and defaults to *t*_1_.

When *t = t*_3_, we obtain the following:(38)$$x\left(t_{3}\right)=e^{A\left(t_{3}-t_{1}\right)}x\left(t_{1}\right)+a_{p}\int_{t_{st}}^{t_{1}}\left\{e^{A\left(t_{1}-\xi \right)}B\left(\xi \right)\left[x\left(\xi \right)-x\left(\xi -T\right)\right]\right\}d\xi$$

### 4.2. Numerical Algorithm

The Lagrange polynomial denoted by *P_L_*(*x*) [16,24] can be used to approximate *f*(*x*) as follows:(39)$$P_{L}\left(x\right)=\sum_{k=0}^{L}f\left(x_{k}\right)\eta_{L,k}\left(x\right)$$
where
(40)$$\eta_{L,k}\left(x\right)=\prod_{\begin{array}{c} 0\leq m\leq k\\ m\neq L \end{array}}\frac{x-x_{m}}{x_{L}-x_{m}}$$

It is worth noting that *x_i_*, *x_i_*_+1_, and *x_i_*_+2_ only exist when *L* = 2, and then, the Simpson’s 3/8 formula (41) can be derived using *P*_2_(*x*).
(41)$$\int_{x_{i}}^{x_{i+2}}f\left(x\right)dx=\frac{h}{3}\left(f\left(x_{i}\right)+4f\left(x_{i+1}\right)+f\left(x_{i+2}\right)\right)$$

Based on (41), (38) can be rewritten as follows:(42)$$\begin{array}{ll} x\left(t_{3}\right)= & e^{2Ah}x\left(t_{1}\right)+\frac{a_{p}h}{3}\{e^{2Ah}B\left(t_{1}\right)\left[x\left(t_{1}\right)-x\left(t_{1}-T\right)\right]+4e^{Ah}B\left(t_{2}\right)\left[x\left(t_{2}\right)-x\left(t_{2}-T\right)\right]\\ & +B\left(t_{3}\right)\left[x\left(t_{3}\right)-x\left(t_{3}-T\right)\right]\} \end{array}$$
which can be expressed as follows:(43)$$\begin{array}{ll} x\left(t_{i+2}\right) & =e^{2Ah}x\left(t_{i}\right)+\frac{a_{p}h}{3}\{e^{2Ah}B\left(t_{i}\right)\left[x\left(t_{i}\right)-x\left(t_{i}-T\right)\right]+4e^{Ah}B\left(t_{i+1}\right)\left[x\left(t_{i+1}\right)-x\left(t_{i+1}-T\right)\right]\\ & +B\left(t_{i+2}\right)\left[x\left(t_{i+2}\right)-x\left(t_{i+2}-T\right)\right]\} \end{array}$$

As **B**(*t_i+_*_2_) cannot be solved directly when *i* = *n*, the classical numerical integration method is introduced to determine **x**(*t_n_*_+1_) as follows:(44)$$x\left(t_{n+1}\right)=e^{Ah}x\left(t_{n}\right)+\frac{a_{p}h}{2}\left\{e^{Ah}B\left(t_{n}\right)\left[x\left(t_{n}\right)-x\left(t_{n}-T\right)\right]+B\left(t_{n+1}\right)\left[x\left(t_{n+1}\right)-x\left(t_{n+1}-T\right)\right]\right\}$$

Combining (17), (43), and (44), the transition matrix can be constructed as follows:(45)$$C_{2}={\left[\begin{array}{ccccccc} 0 & 0 & 0 & \cdots & 0 & 0 & 0\\ -e^{2Ah} & 0 & 0 & \cdots & 0 & 0 & 0\\ 0 & -e^{2Ah} & 0 & \cdots & 0 & 0 & 0\\ \vdots & \vdots & \vdots & \ddots & \vdots & \vdots & \vdots \\ 0 & 0 & 0 & \cdots & 0 & 0 & 0\\ 0 & 0 & 0 & \cdots & -e^{2Ah} & 0 & 0\\ 0 & 0 & 0 & \cdots & 0 & -e^{Ah} & 0 \end{array}\right]}_{\left(2n+2\right)\times \left(2n+2\right)}$$
(46)$$D_{2}={\left[\begin{array}{ccccccc} 0 & 0 & 0 & \cdots & 0 & 0 & 0\\ -e^{2Ah}B_{1} & -4e^{Ah}B_{2} & -B_{3} & \cdots & 0 & 0 & 0\\ 0 & -e^{2Ah}B_{2} & -4e^{Ah}B_{3} & \cdots & 0 & 0 & 0\\ \vdots & \vdots & \vdots & \ddots & \vdots & \vdots & \vdots \\ 0 & 0 & 0 & \cdots & -4e^{Ah}B_{i-1} & -B_{i} & 0\\ 0 & 0 & 0 & \cdots & -e^{2Ah}B_{i-1} & -4e^{Ah}B_{i} & -B_{i+1}\\ 0 & 0 & 0 & \cdots & 0 & \frac{3}{2}e^{Ah}B_{i} & \frac{3}{2}B_{i+1} \end{array}\right]}_{\left(2n+2\right)\times \left(2n+2\right)}$$
(47)$$E_{2}={\left[\begin{array}{ccccc} 0 & 0 & \cdots & 0 & e^{At_{f}}\\ 0 & 0 & \cdots & 0 & 0\\ \vdots & \vdots & \ddots & \vdots & \vdots \\ 0 & 0 & \cdots & 0 & 0\\ 0 & 0 & \cdots & 0 & 0 \end{array}\right]}_{\left(2n+2)\times (2n+2\right)}$$
(48)$$I_{2}={\left[\begin{array}{ccccccccc} 1 & 0 & 0 & 0 & \cdots & 0 & 0 & 0 & 0\\ 0 & 1 & 0 & 0 & \cdots & 0 & 0 & 0 & 0\\ 0 & 0 & 0 & 0 & 1 & 0 & 0 & 0 & 0\\ 0 & 0 & 0 & 0 & \cdots & 1 & 0 & 0 & 0\\ \vdots & \vdots & \vdots & \vdots & \ddots & \vdots & \vdots & \vdots & \vdots \\ 0 & 0 & 0 & 0 & \cdots & 0 & 0 & 1 & 0\\ 0 & 0 & 0 & 0 & \cdots & 0 & 0 & 0 & 1\\ 0 & 0 & 0 & 0 & \cdots & 0 & 0 & 1 & 0\\ 0 & 0 & 0 & 0 & \cdots & 0 & 0 & 0 & 1 \end{array}\right]}_{\left(2n+2\right)\times \left(2n+2\right)}$$

According to (45)–(48), the dynamic mapping of the discrete system can be determined as follows:(49)$$\left(I_{2}+C_{2}+\frac{a_{p}h}{3}D_{2}\right)\left[\begin{array}{c} x\left(t_{1}\right)\\ x\left(t_{2}\right)\\ \vdots \\ x\left(t_{n+1}\right) \end{array}\right]=\left(\frac{a_{p}h}{3}D_{2}+E_{2}\right)\left[\begin{array}{c} x\left(t_{1}-T\right)\\ x\left(t_{2}-T\right)\\ \vdots \\ x\left(t_{n+1}-T\right) \end{array}\right]$$

Therefore, the transition matrix of the dynamic system in a tool pass cycle is obtained as follows:(50)$$\Phi_{2}={\left(I_{2}+C_{2}+\frac{a_{p}h}{3}D_{2}\right)}^{-1}\left(\frac{a_{p}h}{3}D_{2}+E_{2}\right)$$

### 4.3. Simulation and Analysis

#### 4.3.1. Single-DOF Milling Model

A benchmark example of the single-DOF milling model is utilized to validate and analyze S38M. Its state-space expression is introduced as (31), where
(51)$$A=\left[\begin{array}{cc} 0 & 1\\ -\omega_{n}^{2} & -2\zeta \omega_{n} \end{array}\right]$$
(52)$$B(t)=\left[\begin{array}{cc} 0 & 0\\ -\frac{h(t)}{m_{t}} & 0 \end{array}\right]$$

To analyze the convergence rate of S38M, the radial immersion ratio *a*/*D* is set as 1 to avoid intermittent milling. The depths of cut are set as 0.0003 m, 0.0008 m, and 0.0015 m. Figure 7 shows the convergence rate comparisons of the S38M, CCM, SDM, and SEM when the spindle speed is 8000 rpm. It is worth noting that the S38M exhibits the highest convergence rate compared with other chatter analysis methods. According to the enlarged figures on the right, the S38M has a good convergence rate even when the discrete number is large. Additionally, the results show that the convergence rate of the SEM fluctuates evidently, and the SEM is not as stable as the other methods. To further analyze the convergence rate of the S38M, the spindle speed is raised to 10,000 rpm, and the other parameters remain unchanged. Figure 8 shows the resultant convergence rates. It can be seen that these results are consistent with the previous results.

The low-immersion condition can be utilized to verify the stability of the convergence rate [25]. The radial immersion ratio is set as 0.05; that is, *a*/*D* = 0.05. The depths of cut are set as 0.0001 m, 0.0002 m, and 0.0003 m. The spindle speed was set to 5000 rpm. Figure 9 shows the convergence rate comparisons of S38M, CCM, SDM, and SEM under low immersion. The convergence rates arranged from high to low are in the following order: S38M, CCM, SDM, and SEM. Moreover, it is seen that the convergence rate of SEM fluctuated remarkably.

The parameters in Section 3.2 are adopted for the S38M. Table 4 lists the computation time of the S38M. It is seen from the table that the S38M has a high calculation accuracy when *a*/*D* = 1. However, as *a*/*D* decreases, the calculation accuracy of the S38M is not as accurate as that of the CCM. The reason lies in that the interpolation of the S38M, which only uses three points, cannot reach a high calculation accuracy, while the interpolation of the CCM uses five points to interpolate, which made it calculate the smaller interval. The *AMRE* and *MSE* are shown in Figure 10. Notice that the calculation accuracy of the S38M and CCM is higher than those of the SDM and SEM when *a*/*D* = 0.05, and the calculation accuracy of the S38M is higher than those of the CCM, SDM, and SEM when *a*/*D* = 1. The reason for the former lies in that some large fluctuations exist in the S38M. The computational time of the above methods is compared in Figure 11. It is observed from the figure that compared with the SDM and SEM, the computational time can be significantly decreased with the S38M and CCM. Although the computational time of the S38M is almost equal to that of the CCM, the S38M has a higher convergence rate when the radial immersion ratio was large.

As described above, the fluctuations appear in Figure 9. It is found that for the S38M, the convergence rate is often a local maximum when the discrete number is an odd one; the convergence rate is often a local minimum when the discrete number is an even one. To further study this problem, the discrete numbers are set as 30, 40, 50, 60, 70, and 80. The corresponding *AMRE* and *MSE* are plotted in Figure 12. The results show that the S38M attains a better computational accuracy when the discrete number is an even one.

#### 4.3.2. Two-DOF Milling Model

The two-DOF milling mathematical model is expressed as follows [17]:(53)$$\begin{array}{l} \left[\begin{array}{cc} m_{t} & 0\\ 0 & m_{t} \end{array}\right]\left[\begin{array}{c} \ddot{x}\left(t\right)\\ \ddot{y}\left(t\right) \end{array}\right]+\left[\begin{array}{cc} 2m_{t}\zeta \omega_{n} & 0\\ 0 & 2m_{t}\zeta \omega_{n} \end{array}\right]\left[\begin{array}{c} \dot{x}\left(t\right)\\ \dot{y}\left(t\right) \end{array}\right]+\left[\begin{array}{cc} m_{t}\omega_{n}^{2} & 0\\ 0 & m_{t}\omega_{n}^{2} \end{array}\right]\left[\begin{array}{c} x\left(t\right)\\ y\left(t\right) \end{array}\right]\\ =-a_{p}\left[\begin{array}{cc} h_{xx}\left(t\right) & h_{xy}\left(t\right)\\ h_{yx}\left(t\right) & h_{yy}\left(t\right) \end{array}\right]\times \left\{\left[\begin{array}{c} x\left(t\right)\\ y\left(t\right) \end{array}\right]-\left[\begin{array}{c} x\left(t-T\right)\\ y\left(t-T\right) \end{array}\right]\right\} \end{array}$$
where the tool is symmetrical by default. *m_t_*, *ζ*, and *ω_n_* in the *x*-direction are identical to those in the *y*-direction. *h_xx_*(*t*), *h_xy_*(*t*), *h_yx_*(*t*), and *h_yy_*(*t*) are the same as those defined in (3)–(6).

A new state vector is defined as $\dot{r}\left(t\right)={\left[x\left(t\right),y\left(t\right),\dot{x}\left(t\right),\dot{y}\left(t\right)\right]}^{T}$. Then, (53) can be expressed as follows:(54)$$\dot{r}\left(t\right)=Ar\left(t\right)+a_{p}B\left(t\right)\left[r\left(t\right)-r\left(t-T\right)\right]$$
where
(55)$$A=\left[\begin{array}{cccc} 0 & 0 & 1 & 0\\ 0 & 0 & 0 & 1\\ -\omega_{n}^{2} & 0 & -2\zeta \omega_{n} & 0\\ 0 & -\omega_{n}^{2} & 0 & -2\zeta \omega_{n} \end{array}\right]$$
and
(56)$$B=\left[\begin{array}{cccc} 0 & 0 & 0 & 0\\ 0 & 0 & 0 & 0\\ -h_{xx}\left(t\right)/m_{t} & -h_{xy}\left(t\right)/m_{t} & 0 & 0\\ -h_{yx}\left(t\right)/m_{t} & -h_{yy}\left(t\right)/m_{t} & 0 & 0 \end{array}\right]$$

The following deduction is the same as that in Section 4.1.

The system parameters of the two-DOF milling model are identical to those of the single-DOF model. The radial immersion ratios are set as 0.05, 0.5, and 1. The figures shown in Table 5 are obtained over a 100×50-sized grid of the system parameters, that is, the spindle speed and the depth of cut. The thin, black, solid line denotes the reference stability lobe diagrams obtained using the SDM when the discrete number is 200 [26]. We observe that the accuracy of the stability lobe diagrams of the S38M is better than that of SDM. Additionally, the computation time is remarkably saved using the S38M.

## 5. Conclusions

In this study, we focused on the prediction of milling stability using composite Cotes-based and Simpson’s 3/8-based methods. First, the composite Cotes-based method is proposed for preventing chatter in milling processes. The three steps for obtaining the milling stability lobe diagrams are as follows: (1) establish the time-delay differential equation while considering the regenerative effects; (2) determine the transition matrix using the integral equations; (3) analyze the system stability according to the Floquet theory. To measure the calculation accuracy, the *AMRE* and *MSE* are adopted in our work. The results demonstrate that for the single-DOF model, when the discrete number is 40, the *AMRE* and *MSE* can be respectively reduced from 0.076 to 0.041 and from 4.24 × 10^−8^ to 2.62 × 10^−8^, and the calculation time can be reduced from 55.63s to 20.21s by using the proposed composite Cotes-based method. In addition, the Simpson’s 3/8-based method is proposed to further improve the calculation efficiency. The results demonstrate that for the single-DOF model, the proposed Simpson’s 3/8-based method significantly improves the convergence speed while sharing the same computation time with the composite Cotes-based method when the radial immersion ratio is large; for the two-DOF model, the accuracy of the stability lobe diagrams of the Simpson’s 3/8-based method are better than those of the SDM, and the computation time is remarkably saved using the Simpson’s 3/8-based method.

## Figures and Tables

**Figure 1 micromachines-13-00810-f001:**
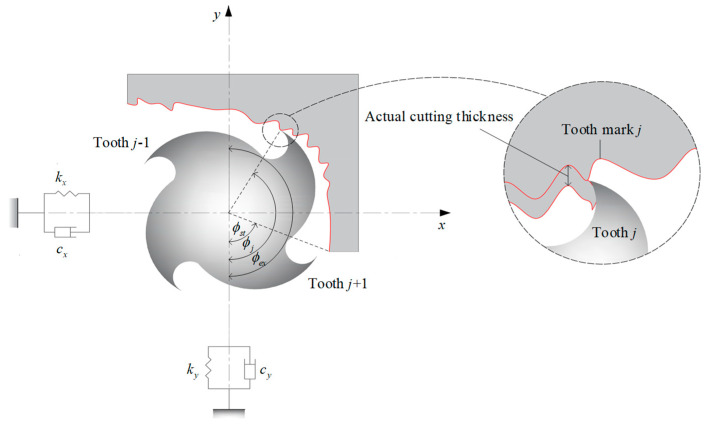
Dynamic system of end-milling with two degrees of freedom.

**Figure 2 micromachines-13-00810-f002:**
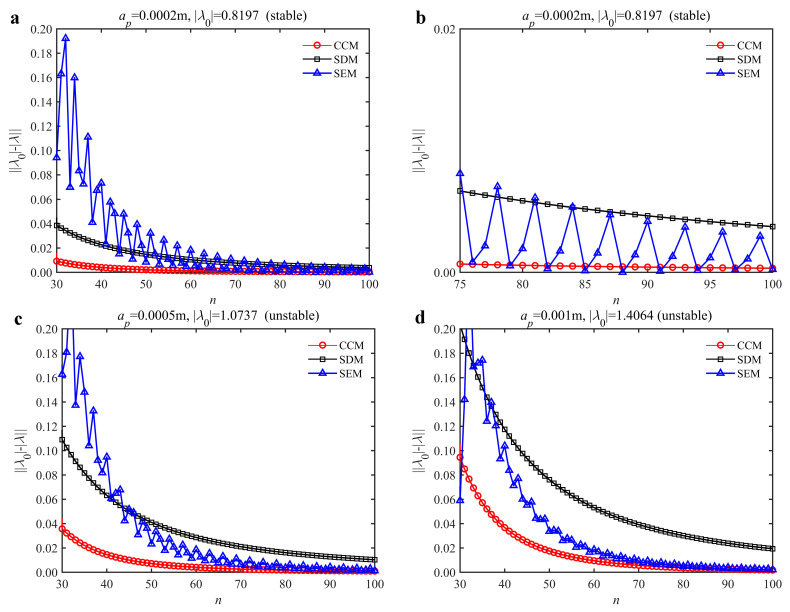
Convergence rate comparisons of the CCM, SDM and SEM when *a*/*D* = 1. (**a**) *a_p_* = 0.0002 m and *n*
∈ [30, 100]; (**b**) *a_p_* = 0.0002 m and *n*
∈ [75, 100]; (**c**) *a_p_* = 0.0005 m and *n*
∈ [30, 100]; (**d**) *a_p_* = 0.001 m and *n*
∈ [30, 100].

**Figure 3 micromachines-13-00810-f003:**
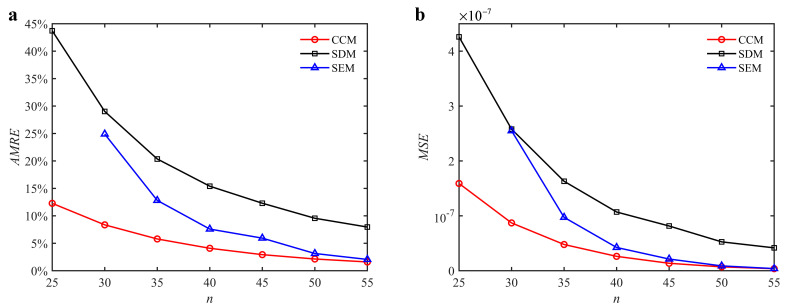
*AMRE* and *MSE* of the CCM, SDM, and SEM when *a*/*D* = 1. (**a**) *AMRE*; (**b**) *MSE*.

**Figure 4 micromachines-13-00810-f004:**
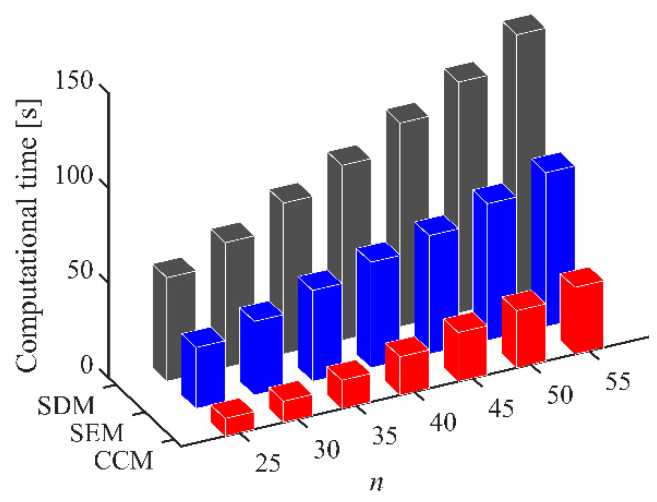
Computational time of the CCM, SDM, and SEM when *a*/*D* = 1.

**Figure 5 micromachines-13-00810-f005:**
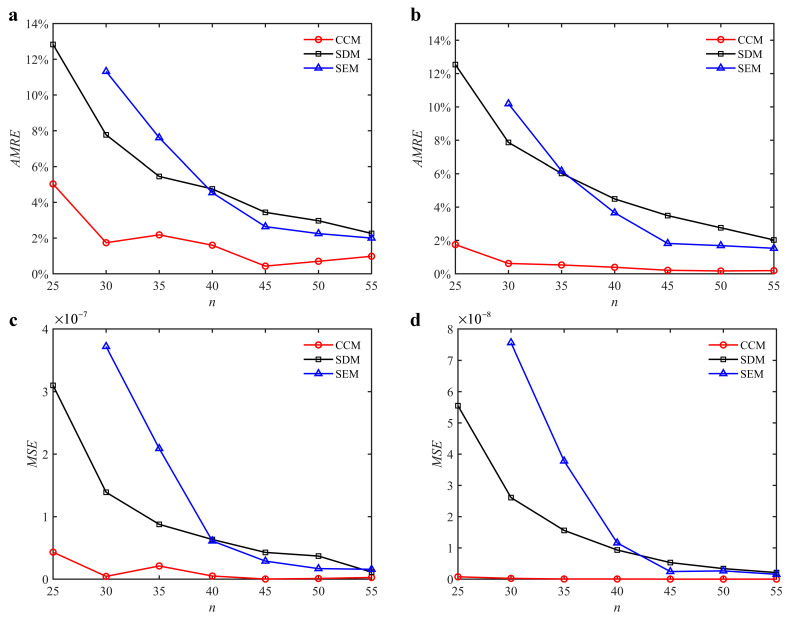
*AMRE* and *MSE* of the CCM, SDM, and SEM. (**a**) *AMRE* when *a*/*D* = 0.05; (**b**) *AMRE* when *a*/*D* = 0.5; (**c**) *MSE* when *a*/*D* = 0.05; (**d**) *MSE* when *a*/*D* = 0.5.

**Figure 6 micromachines-13-00810-f006:**
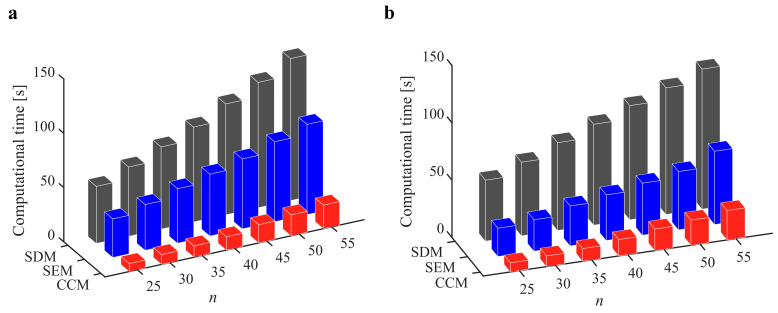
Computational time of the CCM, SDM, and SEM. (**a**) *a*/*D* = 0.05; (**b**) *a*/*D* = 0.5.

**Figure 7 micromachines-13-00810-f007:**
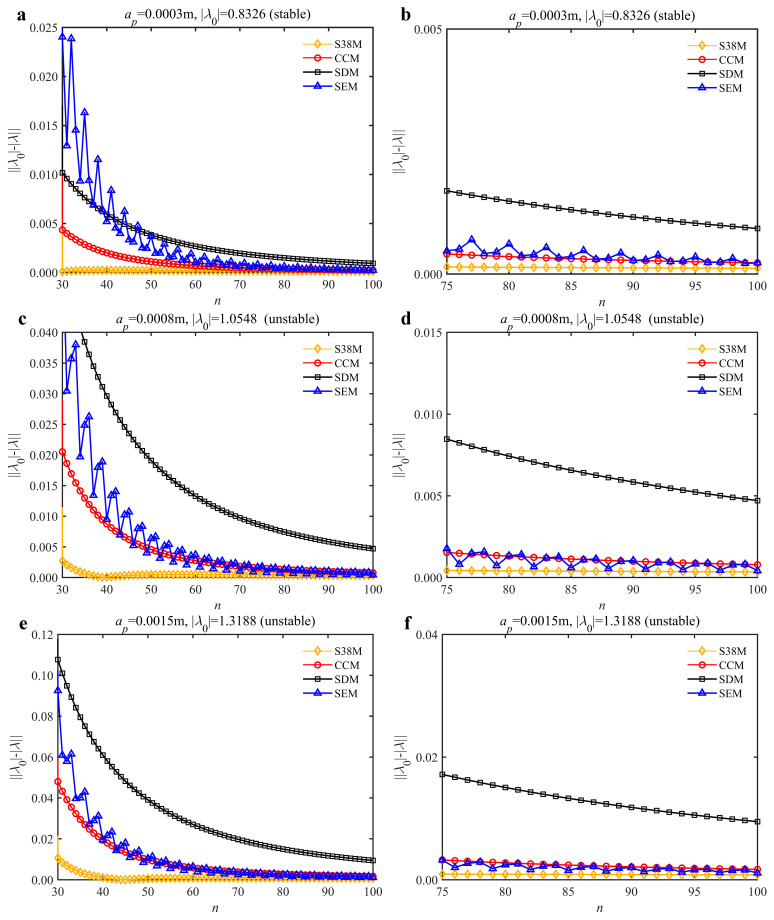
Convergence rate comparisons of the S38M, CCM, SDM, and SEM when *v* = 8000 rpm and *a*/*D* = 1. (**a**) *a_p_* = 0.0003 m and *n*
∈ [30, 100]; (**b**) *a_p_* = 0.0003 m and *n*
∈ [75, 100]; (**c**) *a_p_* = 0.0008 m and *n*
∈ [30, 100]; (**d**) *a_p_* = 0.0008 m and *n*
∈ [75, 100]; (**e**) *a_p_* = 0.0015 m and *n*
∈ [30, 100]; (**f**) *a_p_* = 0.0015 m and *n*
∈ [75, 100].

**Figure 8 micromachines-13-00810-f008:**
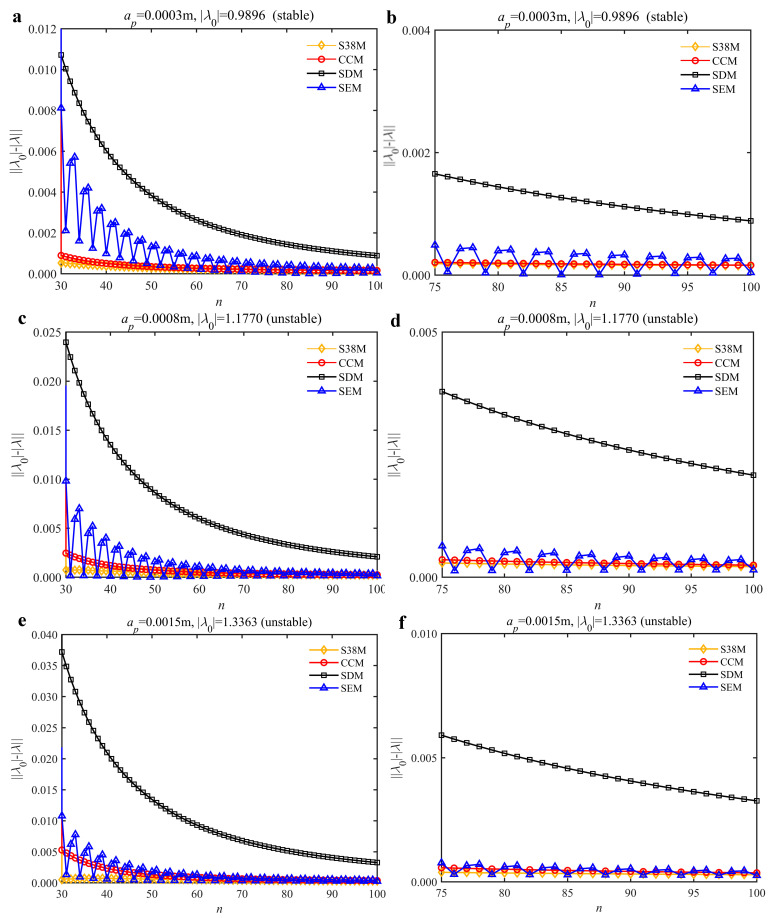
Convergence rate comparisons of the S38M, CCM, SDM, and SEM when *v* = 10,000 rpm and *a*/*D* = 1. (**a**) *a_p_* = 0.0003 m and *n*
∈ [30, 100]; (**b**) *a_p_* = 0.0003 m and *n*
∈ [75, 100]; (**c**) *a_p_* = 0.0008 m and *n*
∈ [30, 100]; (**d**) *a_p_* = 0.0008 m and *n*
∈ [75, 100]; (**e**) *a_p_* = 0.0015 m and *n*
∈ [30, 100]; (**f**) *a_p_* = 0.0015 m and *n*
∈ [75, 100].

**Figure 9 micromachines-13-00810-f009:**
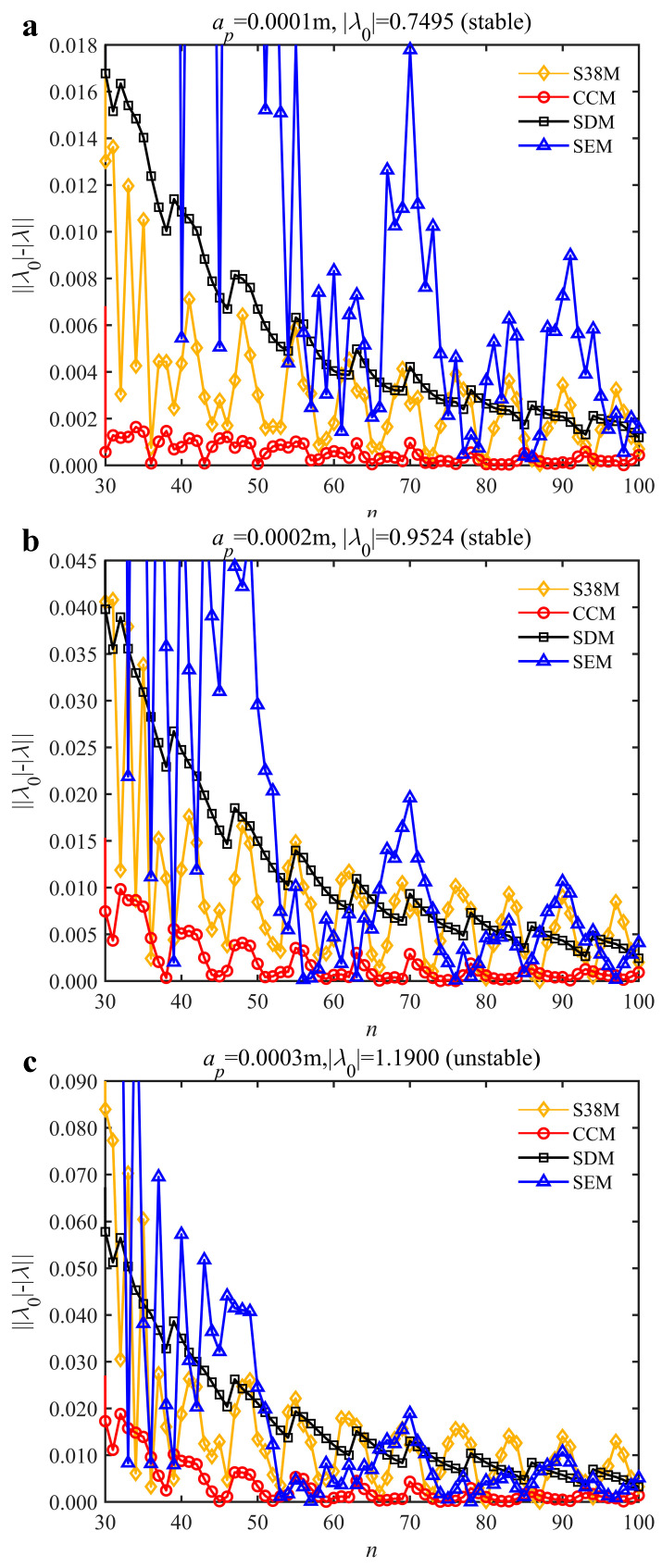
Convergence rate comparisons of the S38M, CCM, SDM, and SEM when and *a*/*D* = 0.05. (**a**) *a_p_* = 0.0001 m; (**b**) *a_p_* = 0.0002 m; (**c**) *a_p_* = 0.0003 m.

**Figure 10 micromachines-13-00810-f010:**
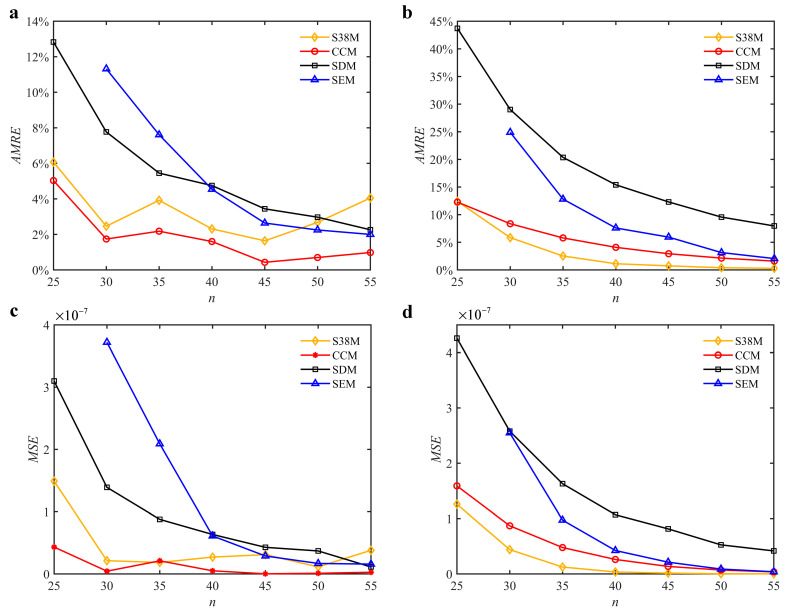
*AMRE* and *MSE* of the S38M, CCM, SDM, and SEM. (**a**) *AMRE* when *a*/*D* = 0.05; (**b**) *AMRE* when *a*/*D* = 1; (**c**) *MSE* when *a*/*D* = 0.05; (**d**) *MSE* when *a*/*D* = 1.

**Figure 11 micromachines-13-00810-f011:**
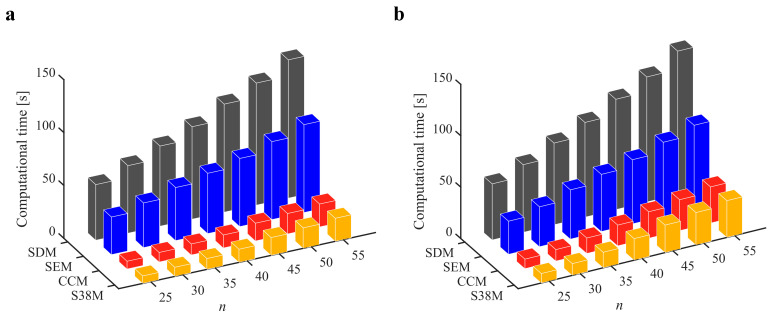
Computational time of the S38M, CCM, SDM, and SEM. (**a**) *a*/*D* = 0.05; (**b**) *a*/*D* = 1.

**Figure 12 micromachines-13-00810-f012:**
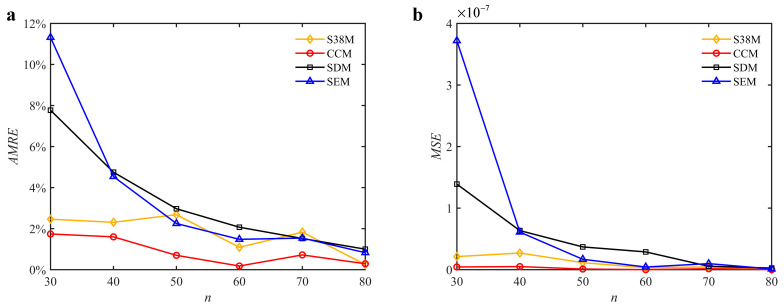
*AMRE* and *MSE* of the S38M for the fluctuation analysis when *a*/*D* = 0.05. (**a**) *AMRE*; (**b**) *MSE*.

**Table 1 micromachines-13-00810-t001:** Stability lobe diagrams of the CCM, SDM, and SEM when *a*/*D* = 1.

*n*	25	40	55
CCM	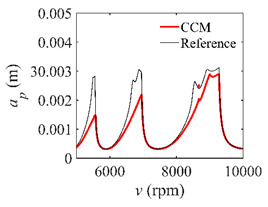	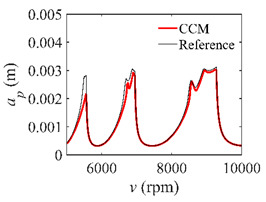	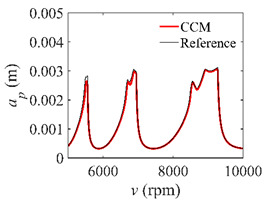
SDM	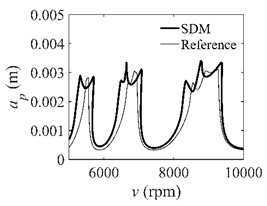	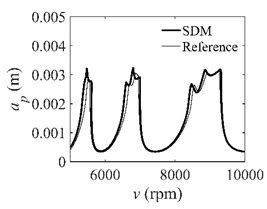	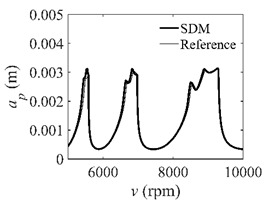
SEM	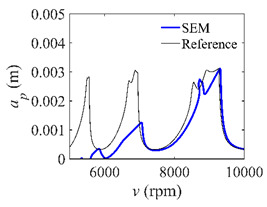	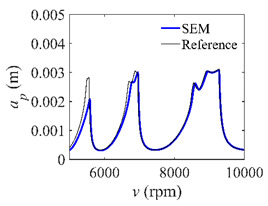	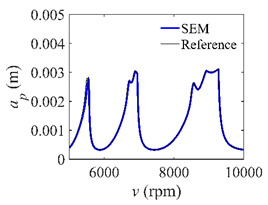

**Table 2 micromachines-13-00810-t002:** Stability lobe diagrams of the CCM, SDM, and SEM when *a*/*D* = 0.05.

*n*	25	40	55
CCM	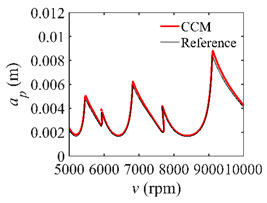	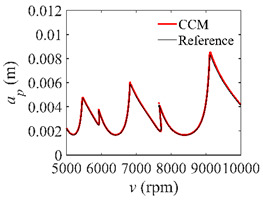	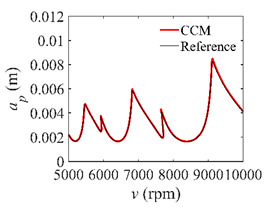
SDM	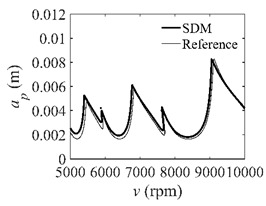	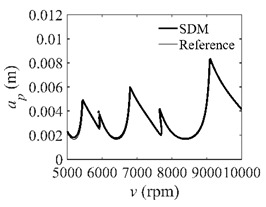	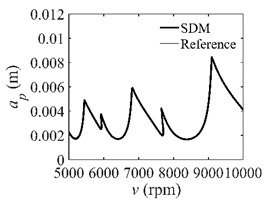
SEM	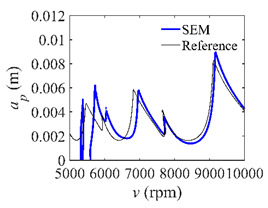	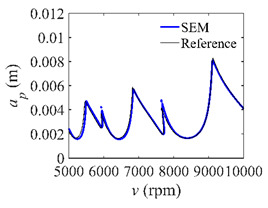	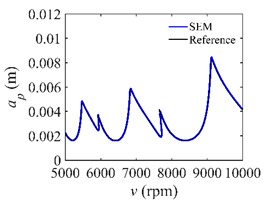

**Table 3 micromachines-13-00810-t003:** Stability lobe diagrams of the CCM, SDM, and SEM when *a*/*D* = 0.5.

*n*	25	40	55
CCM	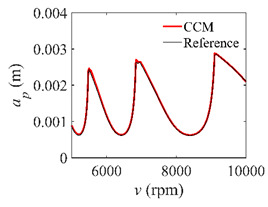	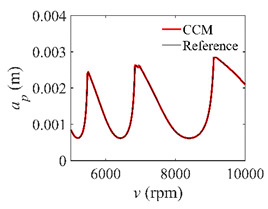	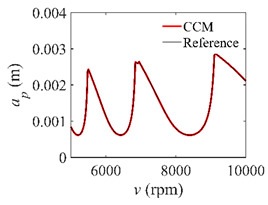
SDM	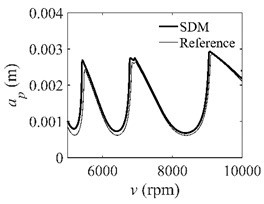	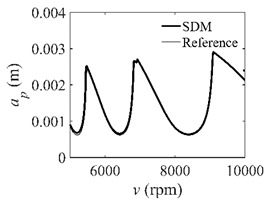	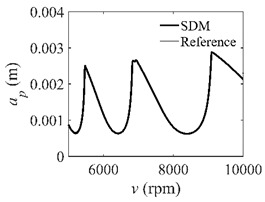
SEM	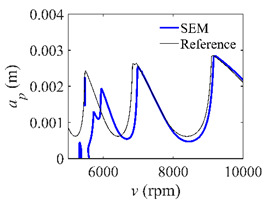	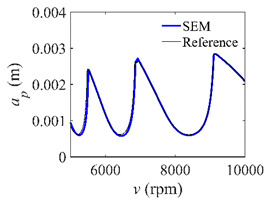	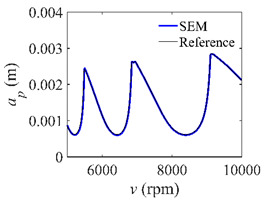

**Table 4 micromachines-13-00810-t004:** Stability lobe diagrams of the S38M.

*n*	25	40	55
*a*/*D* = 0.05	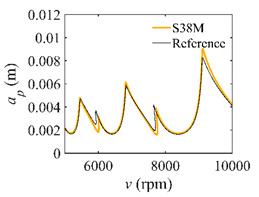	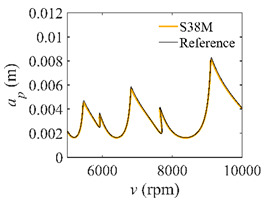	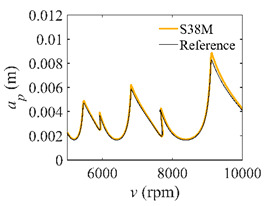
*a*/*D* = 0.5	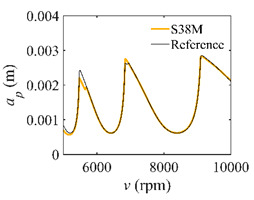	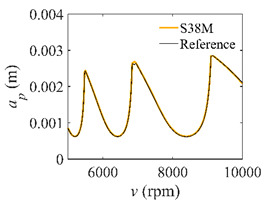	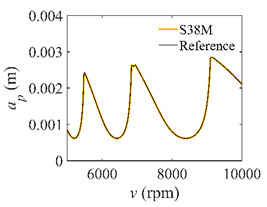
*a*/*D* = 1	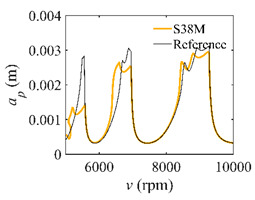	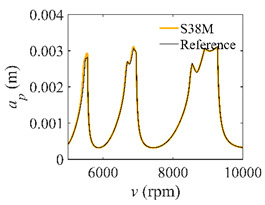	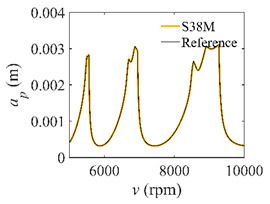

**Table 5 micromachines-13-00810-t005:** Comparisons of S38M and SDM for the two-DOF milling model.

	S38M	SDM
*a*/*D* = 0.05	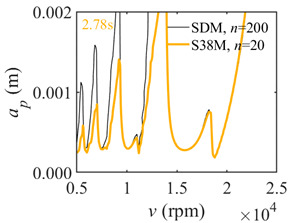	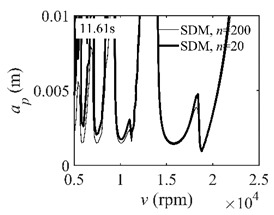
*a*/*D* = 0.5	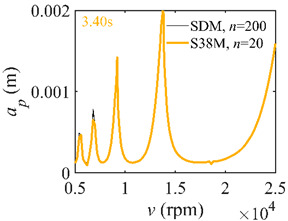	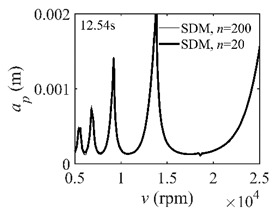
*a*/*D* = 1	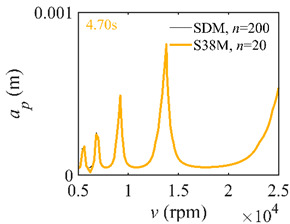	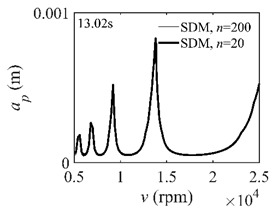

## Data Availability

The data presented in this study are available on request from the corresponding author.

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
