# Peer review of "Predicting Milling Stability Based on Composite Cotes-Based and Simpson’s 3/8-Based Methods"

_micromachines, 2022, doi:10.3390/mi13050810_

Round 1

Reviewer 1 Report

Effective solution of machining stability in the time domain via various numerical integration schemes is a popular topic in literature, but an important one from a practical point of view, because there are a variety of relatively complex tool geometries on the market for which the simpler ZOA method is not sufficient.  The proposed method shows good convergence and apparently very good time performance compared to the other methods. However, I would have a reservation about the presented comparison of time and accuracy. In Figure 3 the method S38M does not seem to be more accurate than SDM as claimed - here I would expect a metric for the accuracy of the solution to be introduced in order to compare the time consumption rigorously. 

Author Response

Thank you very much for your comments.  As the requirements of the reviewer, we have revised the questions raised by the reviewer one by one.

Reviewer 2 Report

The work is devoted to the mathematical analysis of oscillations in the milling process. Currently, a very large number of works have been published on this topic, including those with very detailed experimental studies. The authors present only the results of mathematical calculations, which reduces the relevance of the presented results.

The design is generally well done, but some changes need to be made.

The calculation methodology is adequate and modern.

The work is replete with formulas, but the description of the calculations looks bad. 5-10 lines of text on separate graphs do not reveal the results obtained in sufficient detail.

Notes:

  1. First, the text should go, and only then the drawing. (picture 1)
  2. only 9 publications in the last 5 years
  3. Figures 2-5 do not have captions like a-d
  4. Tables 1-3 look very bad. Better to just make drawings with captions.
  5. The specific purpose of the work is not indicated. Do the authors simply compare methods for calculating fluctuations in a technological system?
  6. There is no discussion of the results. The authors do not give any reasoning on the topic of the results obtained. There is only a description of the equations and the graphs derived from them.
  7. There is no proof of performance for the given model in the work.
  8. The conclusions are written in too general terms. There is no specifics: how much the accuracy has increased, how much the speed of calculations has changed, etc.
  9. Check the link [14].

Based on the totality of the comments, I believe that the article requires significant revision.

Author Response

(The authors gave the same response as above.)

Round 2

Reviewer 2 Report

The authors took into account the comments and made the necessary changes to the work. Answers provide detailed answers and fixes.

I believe that in its current form the article can be accepted for publication.